# Track-Index-Guided Sustainable Off-Road Operations Using Visual Analytics, Image Intelligence and Optimal Delineation of Track Features

Manoj Kumar Kalra [1,2], Sanjay Kumar Shukla [2,3] and Ashutosh Trivedi [2,*]

1 Defence Geoinformatics Research Establishment (DGRE), DRDO, Chandigarh 160036, India
2 Department of Civil Engineering, Delhi Technological University, Delhi 110042, India
3 Discipline of Civil Engineering, School of Engineering, Edith Cowan University, Perth, WA 6027, Australia
* Correspondence: atrivedi@dce.ac.in

**Abstract:** Visual-analytics-guided systems are replacing human efforts today. In many applications, movement in off-road terrain is required. Considering the need to negotiate various soft ground and desertic conditions, the beaten tracks of leading vehicles considered to be safe and suitable for guiding are used in such operations. During night, often, these tracks pass through low-contrast conditions posing difficulty in their identification. The maximization of track contrast is therefore desired. Many contrast enhancement techniques exist but their effectiveness varies as per the surrounding. Other than conventional techniques, the role of texture too becomes important for enhancing the differentiable track contrast. Gray-level co-occurrence matrix (GLCM)-based statistic measures are used here to evaluate the track texture. These measures are seen to improve the contrast of vehicle tracks significantly. A track-index-based technique is proposed to sort various images as per their effectiveness in increasing the track contrast. Different forms of track indices are proposed and compared. The proposed track index is seen as effective in sorting 88.8% of contrast images correctly. The proposed technique of creating and sorting images based on the contrast level is seen as a useful tool for improved fidelity in many difficult situations for making the off-road operations sustainable.

**Keywords:** texture; GLCM; rut; vehicle tracks; off-road; unpaved; image contrast; on-board; sustainable operations





## 1. Introduction

Vehicular movement in off-road unpaved areas is a common requirement for applications, particularly in defense, forestry, agriculture and unmanned ground vehicles. During operations, the vehicles at time need to pass through many of the soft ground conditions. The beaten tracks of leading vehicles, reported as being paths which are safe and suitable for guiding, are followed at times. At times, the track impressions of the leading vehicle need to be followed for strategic reasons [1]. During night time operations, low-contrast conditions are common hindrances for these vehicles. Moreover, these days, visual-analytics-guided systems are replacing human efforts. Vision-based systems are increasingly being used in many such manned and autonomous ground vehicles [2].

Investigators conducted a study of the rut following robotic movement on unpaved terrain [3]. Monocular-camera-based off-road track detection for the path following robot movement is proposed [4]. In comparison to on-road surfaces or lane classification, off-road scenarios are shown to face many challenging situations. There are no well-defined edge cues and the tracks pass through a diversity of natural terrain surfaces. A review of the traversable path for autonomous ground vehicles in off-road detection is reported [5].

The vehicle tracks captured by these cameras have limited contrast. The changed illumination conditions, cluttered backgrounds, wetness and so forth bring about great challenges in the identification of tracks. In order to make these operations sustainable in

such scenarios, it is important to look into alternate means too that can improve the track contrast in a given situation.

A deep-learning-based CNN method is presented for lane detection using vision cameras [6]. The use of the generative adversarial network (GAN) for addressing the issue of extracting road boundaries in complex terrain scenarios is presented [7]. A framework for combined road tracking for paved roads and dirt roads is given [8]. In this work, a CNN-based measurement utilizing the self-similarity of (dirt) road areas is shown to be tracked with a lookahead length of 25 m. Although the machine learning aspect is important for track detection, even for these studies to accurately mark the vehicle tracks with feeble boundaries, a robust and accurate dataset is needed.

Furthermore, the role of traditional image processing and computer vision techniques is important here. In the context of 3D robot vision, it is seen that combining both linear subspace methods and deep convolutional prediction achieves improved performance along with several-orders-of-magnitude-faster runtime performance compared to the state of the art [9]. Ten different concerns for deep learning are seen and it is suggested that deep learning must be supplemented by other techniques if we are to reach artificial general intelligence [10]. In the study of microcirculation images, the limitations of deep learning are reported and a hybrid model to strike a balance between accuracy and speed by combining traditional computer vision algorithms and CNN is proposed [11]. Furthermore, for these advanced algorithms too, the correct delineation of tracks in different scenarios is required. This is where the traditional image enhancement techniques play a role.

Several techniques for image enhancement and better contrast are discussed [12,13]. These techniques are primarily based on filters and histogram stretching. Several techniques of image enhancement have been compiled [14].

The unpaved track features which look like edges in coarse-resolution images take the shape of elongated areas in fine-resolution images. This additional aspect of comparative change in the texture of tracks with respect to their surroundings can reveal useful information for the improved interpretation of track features.

There are various techniques of texture estimation; however, the GLCM-based approach employing various statistical measures has shown very good results in a variety of applications [15]. The relationship between the pixels in the image is characterized by using different statistical measures such as contrast, energy, entropy, homogeneity, etc. Texture analysis using GLCM is employed in the detection of road boundaries [16]. The contrast of tracks using texture based measures depends upon the surrounding terrain features. A study on the various aspects influencing the texture is proposed [17].

Considering various measures for an enhancement in track contrast, an attempt is made to quantify the effectiveness in a given surrounding track contrast using a new track index (TI) and to sort the images as per the track contrast. In this paper, different aspects related to contrast enhancement and the optimization aspects are presented.

The structure of this paper is as follows: Firstly, the introduction discusses the background about the problem statement, and various ongoing attempts related to the study, the gaps and the brief about the proposed solution are given in Section 1. The related work to the proposed study is then given in Section 2. The tools and methodology used for image contrast enhancement, both in terms of conventional and texture-based measures, are discussed in Section 3. The result of the application of various image enhancement measures and the proposed track index for the quantification of the track contrast using various track indices are given in Section 4. The section also discusses the result of the comparative analysis of the track contrast data of image analysis and the details for sorting the images based on track contrast. Section 5 contains the discussion about the result of study and the associated aspects. Section 6 contains the conclusion about the study, summarizing its role in many difficult situations for making off-road operations sustainable.

## 2. Review of Past Works

The vehicular movements in off-road areas pose many challenging situations which are essential to be addressed for sustainable operations. In places, the ground strength gives way in different environmental conditions, thereby needing some strengthening measures. Situations of low contrast are other issues which exert challenges on decision making for the track or rut following vehicular operations. Considerable efforts continue to make the off-road operations sustainable from varied perspectives. Improving the strength of unpaved terrain using geosynthetics [18] and its evaluation [19] for sustaining vehicular loads are a few such alternatives employed. Machine learning processes as employed in the crack detection of bridges and asphalt [20] and the proposed self-attention-based U-net model [21] can be extended for the autonomous detection of track features too. The delineation of track zones in spatially varying, low-contrast terrain is another important issue that requires attention for sustainable operations.

Tracks are seen to be distinguishable from their surroundings not only by the variation in tone but also by the pattern and texture which are differentiable with respect to their surroundings. Investigators of [22] used a mobile robot with a vision system and used an artificial neural network (ANN) for real-time terrain characterization based on its traversability. An algorithm [23] was introduced for a line follower robot to achieve the ability to autonomously follow a path that had straight lines.

Image stretching, power functions, low–high-pass filters, histogram stretching and its equalization are some of the well-known techniques employed for image enhancement [12]. A framework based on multipeak-mean-based optimized histogram modification was introduced to demonstrate the enhancement in contrast [24]. Edge detection techniques have been used in preserving the high-frequency components and structural features for the detection of linear features. Broadly, the edge detection algorithms are grouped into two types on the basis of derivatives [25]: (a) gradient-based operators which compute the first-order derivatives of an image and (b) Laplacian-based operators which are based on the second-order derivatives of an image. Both gradient and Laplacian filters are used to highlight discontinuities in an image. Using these in the background, many variants of the edge detection algorithms have been developed; Sobel, Prewitt, Canny, LoG, etc., have been studied for a comparison of results, and each of them have been shown to have their own merits.

Texture representing the relation of the pixels with reference to their neighborhood reveals very useful information distinctive to the object. Texture-based analysis has been employed for image understanding and for different applications by various researchers. Various approaches have been used to describe the textures in the images which differ from each other mainly by the method used for extracting textural features. These approaches which are based on four methods, viz. statistical methods, structural methods, model-based methods and transform-based methods, have been compared [26]. Several applications make use of texture information for required feature extraction. A survey of texture feature extraction methods is given in [27].

In content-based image retrieval, a combination of edge information and texture information using co-occurrence matrix properties is used [28]. Texture features are used for evaluating the real-time distress conditions of roads [29]. GLCM texture features are used as feature descriptors for image retrieval in different applications [30]. A local second-order texture entropy to represent the nature of gray-scale variation has been employed, and the authors of [31] proposed an edge detection method based on local texture entropy for better edge detection.

All of these studies are significant from the perspective of enhancing the track contrast, and related tasks are used in this study.

## 3. Tools and Methodology Used

In order to investigate the role of different resolutions, the images of Google Earth at different resolutions were taken as the basis in this study. The images taken for the analysis

were from an area near Chandigarh. The test sites presented in this study were taken based on ground trials conducted in places with desertic terrain features near Suratgarh, Rajasthan, and with alluvial terrain features in Roorkee, Uttranchal, in India. In another set of images of tracks at ground level, vision cameras were used. The analysis was carried out using Sentinel Application Platform (SNAP) 8.0 and MATLAB 2020a software. The following points give details about the methodology used in this study:

### 3.1. Using Some Linear and Non-Linear Transformation Functions

Due to changed environmental conditions, many times, it is not possible to interpret features directly from images captured by cameras. Many features in the image become prominently clear when certain image processing techniques are applied on the images. There are various measures which can be used to enhance the track contrast, and some of which that are relevant to this study are described here.

The contrast between the two pixels is based on the difference between their gray levels. Many times, the full dynamic range is not used in the image, thereby making the image have reduced contrast. Contrast stretching tries to make use of the full dynamic range and improves the contrast uniformly for the whole image.

Contrast stretching as defined as improving the contrast by stretching the range of intensity values in a given range to the desired values. It is defined as follows:

$$g(x,y) = \frac{L-1}{f_{\max} - f_{\min}}(f(x,y) - f_{\min}) \tag{1}$$

where $g(x,y)$ is the array of pixels in the transformed image and $f(x,y)$ is that for the original image; $f_{\max}$ and $f_{\min}$ are the maximum and minimum gray values of the image pixels. $L$ indicates the quantization levels, for instance, an 8-bit image contains $2^8 = 256$ levels. The transformation here can also be made for a given range of gray values using a piecewise linear stretching function.

There is a possibility of improving the image contrast in a specific range of gray levels using various non-linear transformation functions [12,13]. Some of the transformation functions include logarithmic stretching, which enhances the contrast of pixels in a dark region, whereas the reverse function antilog enhances the contrast between bright pixels. Logarithmic stretching is defined using

$$s = c.\log(1+r) \tag{2}$$

where $s$ and $r$ are the output and input pixel values respectively. The parameter $c$ is the scaling constant to obtain the output value in a desired dynamic range.

The power or Gamma function can also be used to carry out the image stretching of the pixels to a varying degree using the following:

$$s = c.r^{\gamma} \tag{3}$$

In this transformation, the parameter $c$ is the scaling constant and the value of $\gamma < 1$ is used when we are more sensitive to changes in the dark as compared to bright areas in the image. Similarly, $\gamma > 1$ is used when we are more sensitive to changes in bright areas than in dark areas.

Another method of increasing image contrast is by manipulating the histogram of an image. A histogram is created by counting the number of times each gray-level value occurs in the image.

$$h(r_k) = n_k \tag{4}$$

where $r_k$ is the $k$th gray level in the range $[0, L-1]$ and $n_k$ is the number of pixels in the image with a gray level of $r_k$. The histogram is normalized by dividing the numbers by the total number of pixels in the image to create a distribution of the percentage of each gray level in the image. The histogram equalization is the most common technique used

for image contrast enhancement. It accomplishes this by effectively spreading out the most frequent intensity values, i.e., stretching out the intensity range of the image. This allows for areas of a lower local contrast to gain a higher contrast. This method usually increases the global contrast of images, whereas another method of adaptive histogram equalization is used for contrast stretching over local areas. In this, the histograms are created for distinct sections of the image and the gray values are re-distributed.

### 3.2. Using Spatial Filters

Some spatial filtering methods that are used to sharpen edges and remove much of the image blur which are quite relevant for enhancing image contrast are used here. In all of these operations, the convolution of images with various filters is carried out using operations such as the following:

$$Conv(w, f) = w(x, y) * f(x, y) = \sum_{s=-a}^{a} \sum_{t=-b}^{b} w(s, t) f(x - s, y - t) \tag{5}$$

Here, *w(x, y)* indicates the filter of size s x t dimension scanned over the image *f(x,y)*. The symbol (*) is used to indicate the convolution of the image and filter. Convolution computes the output based on the weighted average of brightness values of pixels located in a particular spatial frame. The filters or kernel used here was the matrix with values in a given spatial relation, which was used for highlighting a specific feature. The edge filters and various low–high frequency filters were designed based on the above concept.

The boundaries of the track areas represented by edges can be used to differentiate the track area using edge filters. Edges which are a set of connected pixels forming a boundary between two disjointed regions are considered as cues for track and road lane detection [32]. The vehicle tracks that are distinctive from the surroundings at the boundary and the role of edges representing boundaries were thus explored. Edge detection assists in preserving and highlighting the high-frequency components in the image. Edge detection usually depends upon the calculation of first or second derivatives of the image [25]. The first-derivative-based edge filters were designed based on the gradient of the pixel values in the image and were computed as follows:

$$\nabla f = grad(f) = \begin{bmatrix} g_x \\ g_y \end{bmatrix} = \begin{bmatrix} \frac{\partial f}{\partial x} \\ \frac{\partial f}{\partial y} \end{bmatrix} \tag{6}$$

Here, the gradients $g_x$ and $g_y$ are the first derivatives of image *f(x,y)* indicating pixel value changes occurring in both the x and y direction and are represented as the column vector $\nabla f$. The second-derivative-based edge filter was also defined using the Laplacian of the image *f(x,y)*, and was obtained using the second-order differential equation given below:

$$\nabla^2 f = \frac{\partial^2 f}{\partial^2 x} + \frac{\partial^2 f}{\partial^2 y} \tag{7}$$

However, the gradient-based filters were used to highlight the prominent edges, while the Laplacian filters brought out the finer details [33]. Based on the above concept, many of the edge detection filters that are designed include Sobel, Prewitt, Roberts, Canny, etc. These are used to highlight the edges based on some varying concepts, and each of which has its own merits.

The contrast of linear features could also be highlighted by using other high-frequency filters, wherein the low-intensity features are deemphasized. These high-frequency filters which can be useful in differentiating the track and surrounding features based on the frequency of features were used in this study.

### 3.3. Using Texture Measures

The conventional techniques of image enhancement which are primarily based on the general understanding of brightness values in an image assist in highlighting the image features to an extent. When a group of pixels representing any feature is differentiable from its surroundings, the role of texture comes into place. The texture defines the spatial arrangement of these pixels in the feature. With the arrangement of pixels in the track zone being different from its surroundings, the role of texture is therefore explored in this study for improved image intelligence.

The GLCM-based texture measure which is considered as a good descriptor of features was used in this study. It considers the relation between two pixels at a time, called the reference and the neighbor pixel. The concept of measuring texture using GLCM by extracting various texture features is given in [34]. The author introduced fourteen textural features that contain information about image texture characteristics. Later, investigators of [35] identified that only 5 of these 14 measures were sufficient, including energy, entropy, homogeneity, contrast and correlation. The key measures that were used in the current study are described here.

$$Energy = \sum_{i,j} p(i,j)^2 \tag{8}$$

$$Entropy = -\sum_{i,j} (p_{i,j}) \log_2(p_{i,j}) \tag{9}$$

$$Homogeneity = \sum_{i,j} \frac{p(i,j)}{1 + |i - j|} \tag{10}$$

$$Contrast = \sum_{i,j} |i - j|^2 p(i,j) \tag{11}$$

$$Correlation = \sum_{i,j} \frac{(i - \mu_i).(i - \mu_j).p(i,j)}{\sigma_i.\sigma_j} \tag{12}$$

where *p(i,j)* is the probability value recorded for the co-occurrence of cells *i,j* in the GLCM matrix; $\mu_i$ and $\mu_j$ are the means, and $\sigma_i$ and $\sigma_j$ are their standard deviations.

Here, different statistic measures have their own significance, and the details of which are given by the investigators of [34,35]. Energy, represented based on the angular second moment (ASM), measures the textural uniformity, that is, pixel pair repetitions. Entropy measures the disorder or complexity of an image. Homogeneity denotes the absence of intra-regional changes in the image. Contrast is indicative of the spatial frequency of an image and measures the amount of local variations present in the image. Correlation calculates the linear dependency of the gray-level values in the GLCM matrix and indicates how the reference pixel is related to its neighbor.

All of the above statistical measures were used in the analysis of texture over the images, as used in this study.

## 4. Results

In order to understand the effect of different measures in an enhancement in track contrast, images are enhanced using a number of techniques and some of these were used in this study. Since there could be several ways, only some significant measures influencing track contrast were considered here. In order to create the images using various techniques such as edge enhancement, high-pass filters and texture images, a filter of $5 \times 5$ size was convolved over the input images. In the study of texture, GLCM was created by downsizing the quantization levels as the computational complexity of this method is proportional to O ($G^2$) [36]. More levels imply more accurate textural information, but with increased computational cost. Applying a large displacement value to a fine-texture image would yield a GLCM that does not capture detailed textural information [37]. In order to compute

image texture, a horizontal offset of 1 pixel and quantization level of 32 were used to create various texture images.

### 4.1. Effect of Image Enhancement Measures

First of all, a comparison of various image enhancement measures was done on the images of different scales. So, the images of a road junction in an area near Chandigarh were captured at different resolutions, as given in Figure 1a (source: Google Earth, Maxar Technologies).

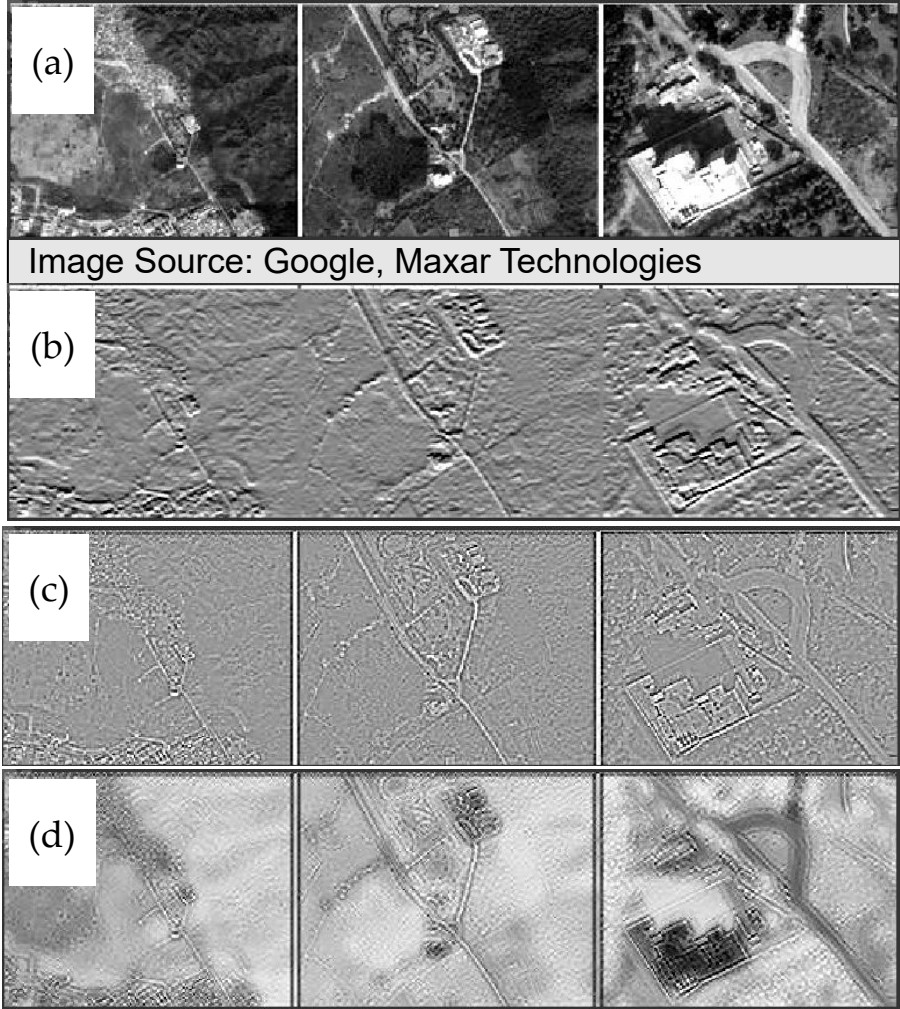

**Figure 1.** Multiscale images of roads: (**a**) original in gray tone indicating coarse-, medium- and fine-scale images (source: Google Earth) enhanced using (**b**) Sobel edge detection filter, (**c**) Laplacian filter and (**d**) high-pass filter. (Images created using SNAP 8.0 software).

Among conventional methods, edge detection filters, using Sobel as the first-order and Laplacian as the second-order derivative, and high-pass filters were used and convolved over the images. The results of the analysis are shown in Figure 1b–d, respectively.

The texture analysis on the same images was also carried out using GLCM. The images representing different statistical measures of texture were created as shown in Figure 2.

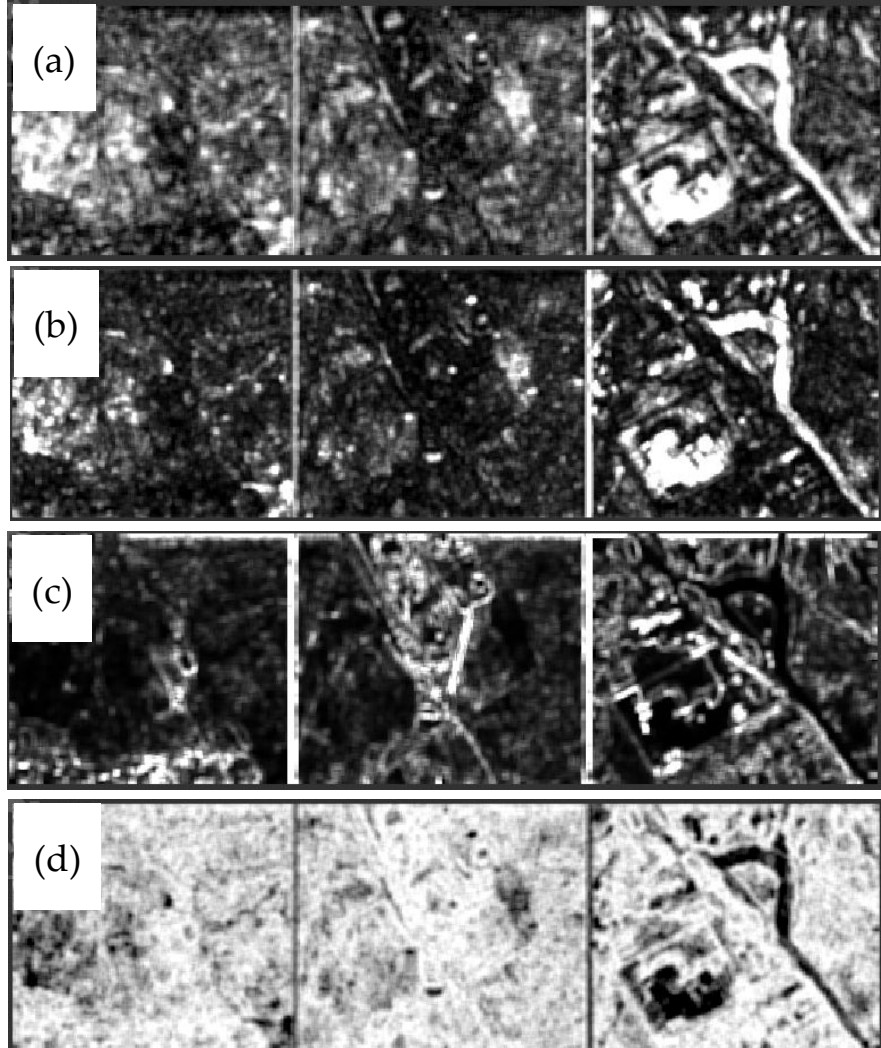

**Figure 2.** Multiscale images of roads indicating coarse-, medium- and fine-scale images enhanced using (**a**) homogeneity, (**b**) energy, (**c**) contrast and (**d**) entropy as the texture measures. (Images created using SNAP 8.0 software).

Figure 2 contains images that represent (a) homogeneity, (b) energy, (c) contrast and (d) entropy as the texture measures. An analysis of the images is shown in Figure 1, and Figure 2 indicates one notable point that as the resolution increases, the role of enhancement measures using the non-linear filters and texture increases. This is because the pixels representing the track features as a group are prominently different to the surroundings.

The effect of various measures was investigated further to identify their role on the ground-scale images containing vehicle tracks. The results of the analysis carried out on the images of track impressions left over by the leading vehicle in desertic tracks are shown in Figure 3.

The influence of other texture measures such as GLCM mean and GLCM variance was also explored here. The visual appearance of the results reveals the importance of texture in delineating the tracks in a better way than the original gray image.

These results, along with standard image enhancement measures, can be used in devising an improved way for differentiating the track zones.

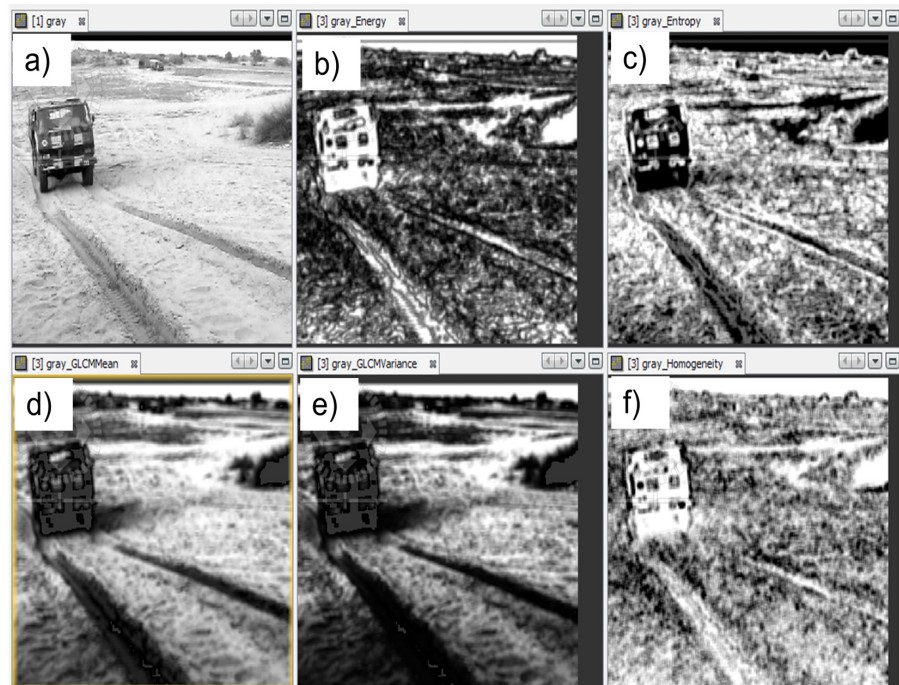

**Figure 3.** Effect of different texture measures on tracks in desertic terrain: (**a**) original gray image and texture images created using filters of (**b**) energy, (**c**) entropy, (**d**) GLCM mean, (**e**) GLCM variance and (**f**) homogeneity image. (Images created using SNAP 8.0 software).

### 4.2. Quantification of Track Contrast Using Proposed Track Indices

In order to compare the track contrast quantitatively, an index-based approach was proposed in this study. First of all, a cross sectional profile was drawn across the track in each image as per the details marked on one of the texture images, as displayed in Figure 4.

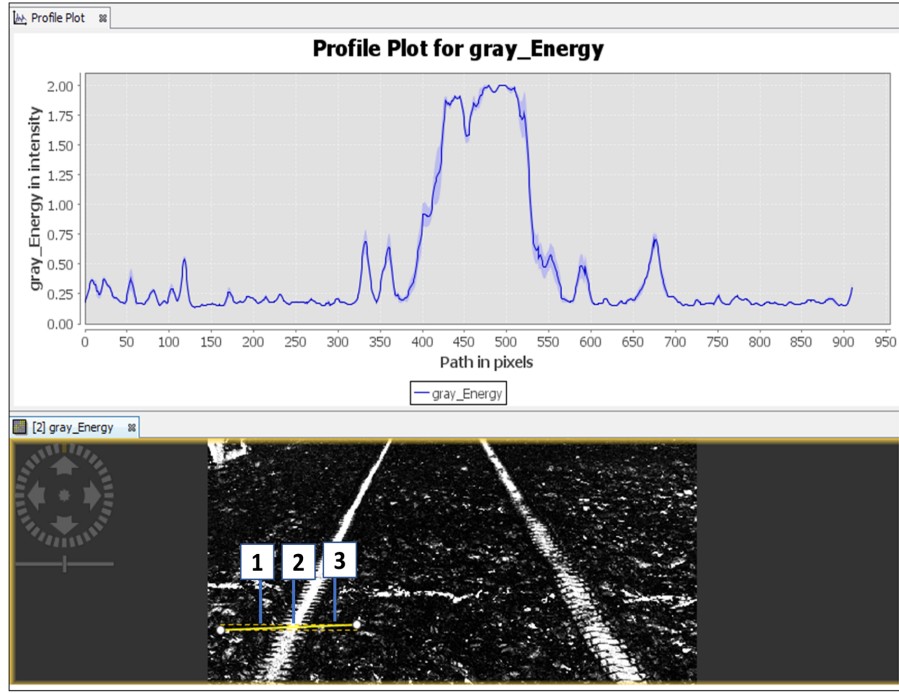

**Figure 4.** Location of pixels chosen for comparing the contrast in track areas with reference to its surroundings (Image created using SNAP 8.0 software).

As the contrast of tracks needs to be considered with respect to its surrounding, for each of the image enhancement measures, areas representing pixels on-track ($P_T$) and pixels off-track ($P_{OT}$) were therefore considered. As local variation in the feature values was also expected, the mean value of the features value was taken. In this study, each measure was averaged over a rectangular area with a width of $11 \times 100$ pixels.

In order to compare the contrast using various measures, the difference in the mean value of the statistical measure ($x$) on the pixels along the track ($P_T$) and in the pixels in the off-track ($P_{OT}$) areas were computed as shown below:

$$P_{OT} = \frac{n_1 x_1 + n_3 x_3}{n_1 + n_3} \tag{13}$$

$$P_T = \frac{n_2 x_2}{n_2} \tag{14}$$

Here, in a given enhancement measure, $x_1$ and $x_3$ are the values at the selected pixels in the off-track zones located to the left and right of the track, respectively. Similarly, $x_2$ indicates the values for pixels on the track.

As the range of values computed for different statistic measures shall be different, in order to compare the two measures, normalization was carried out, as shown below:

$$z = \frac{x - x_{\min}}{x_{\max} - x_{\min}} \tag{15}$$

where $z$ is the normalized value of $x$ data representing the contrast enhancement measure and $x_{\min}$ and $x_{\max}$ are the minimum and maximum values of its range. Here, the range in the numeric values of data gets normalized between 0 and 1.

A number of alternates were considered for defining the contrast quantitatively. The following four measures of track index as defined below were used in this study.

### 4.3. Based on Difference in Mean Values

The track index was defined on the basis of difference in the mean values of pixels on-track and located off-track. Depending upon the used statistic measure, the values of measure could be higher either on-track or off-track, and the absolute difference was thus taken as the measure, as described below:

$$TI(D) = \max(P_{OT}, P_T) - \min(P_{OT}, P_T) \tag{16}$$

(a)  Based on Ratio of Mean Values

The track index here was considered on the basis of the ratio of the mean values of pixels on-track and located off-track. Depending upon the used statistic measure, the values of measure could be higher either on-track or off-track; therefore, the ratio of maximum and minimum values was taken as the measure, as defined below:

$$TI(R) = \frac{\max(P_{OT}, P_T)}{\min(P_{OT}, P_T)} \tag{17}$$

(b)  Based on Normalized Difference in Mean Values

The track index here was considered by normalizing the difference in mean values of pixels on-track and located off-track. The measure was defined as follows:

$$TI(ND) = \frac{\max(P_{OT}, P_T) - \min(P_{OT}, P_T)}{\max(P_{OT}, P_T) + \min(P_{OT}, P_T)} * 100 \tag{18}$$

(c)  Based on the Ratio of Coefficient of Variance

The track index here considered the distinguishing feature of the track based on the standard deviation of the values of pixels on-track and located off-track. The co-efficient of variance (*CV*), which is the ratio of the standard deviation to the mean value, was used here. Considering this measure, the variance-based index was then defined as below:

$$TI(CV) = \frac{\max(CV(P_{OT}, P_T)) - \min(CV((P_{OT}, P_T))}{\max(CV(P_{OT}, P_T)) + \min(CV((P_{OT}, P_T))} * 100 \tag{19}$$

All of the above track indices were evaluated for their effectiveness in this study. In order to do so, these indices were computed using data for different enhancement measures.

### 4.4. Analysis of Track Contrast Data

The track impressions left over by the leading vehicle in the off-road terrain were used in this study. The track images were captured using vision cameras on board the vehicle. The procedure for obtaining the maximum contrast using different contrast enhancement measures is indicated in Figure 5 below.

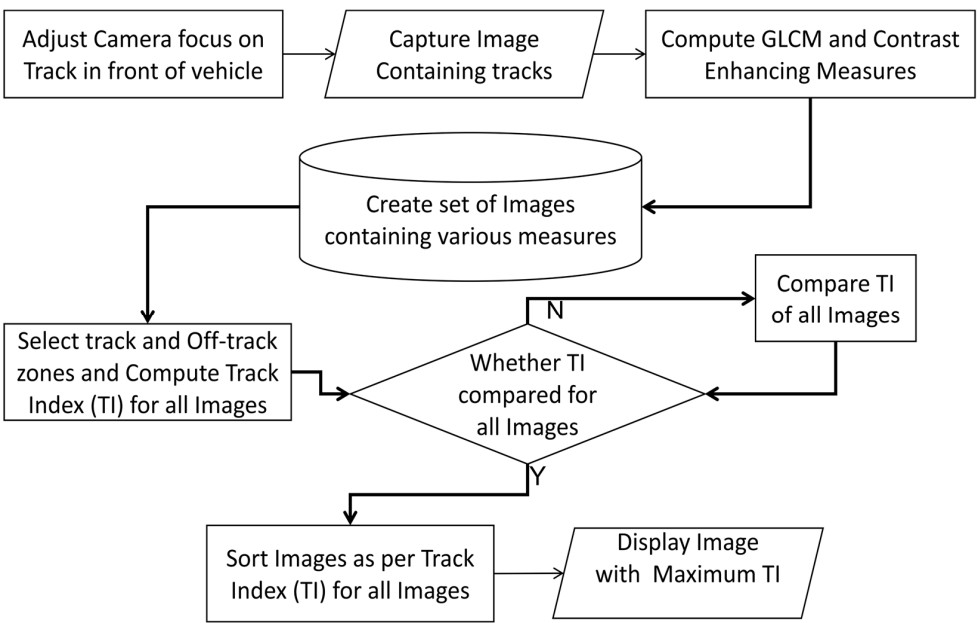

**Figure 5.** Process flow to generate image with maximum track index outside of using various contrast enhancement measures.

This procedure was applied for computing the track indices on various images created using different enhancement measures. The results of analysis using different measures are presented in a consolidated image in Figure 6.

The visual appearance does create an enhancement in the track index via one or another measure. In order to make a comparative analysis, the pixel values were computed across the track area. The mean and standard deviation (sigma) of the pixel values on-track and located off-track were also computed. Further computations were then carried out for evaluating different track indices. The analysis of data was consolidated, as shown in Table 1.

Here, the mean values and standard difference for off-track and on-track pixel values were taken as the basis for evaluating various track indices. These indices indicated the comparative difference in track index values for different image enhancement measures. In order to evaluate the suitability of different measures, a new method was proposed. The images were first sorted and ranked based on the visual appearance. The ranking was then computed for each of the proposed track indexes and marked for each enhancement measure. The obtained details are summarized in Table 2.

The grading of each of the proposed track indexes was then compared with the visually graded rank of the images as per the track contrast. As the manually graded rank was subjected to some uncertainty in the correct ordering, a deviation of two ranks in computed rank from the manually graded rank was considered as acceptable. The table also indicates the results of the acceptable image ranking. The difference from the manual rank in the table is marked with *0* where the rank is within 2 images and marked as *1* where the difference is more than 2 images. The correctness of results was then computed and is given in the table above.

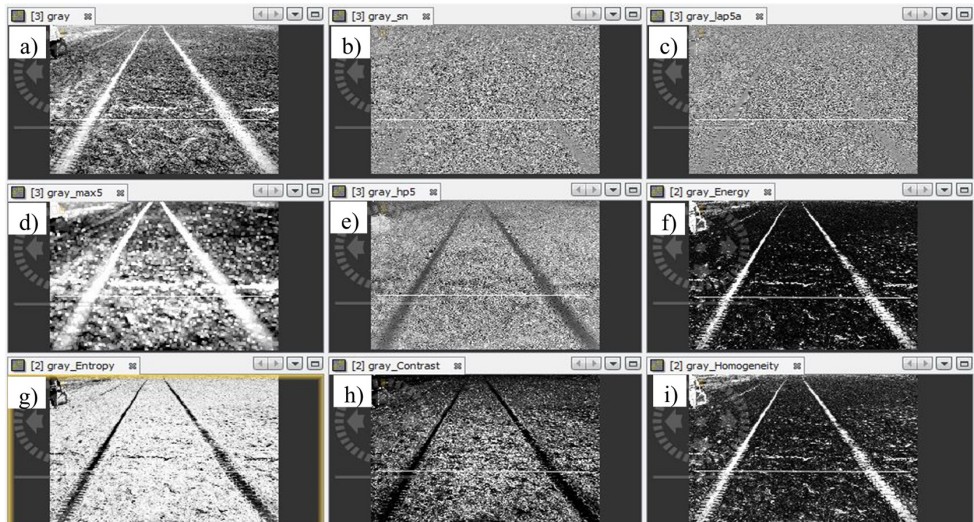

**Figure 6.** Different contrast enhancement measures of leading vehicle tracks: (**a**) original gray image of the tracks and images created using (**b**) Sobel edge detection filter, (**c**) Laplacian edge filter, (**d**) non-linear maximum filter and (**e**) high-pass filter, and using texture measures of (**f**) energy, (**g**) entropy, (**h**) contrast and (**i**) homogeneity. (Images created using SNAP 8.0 software).

**Table 1.** Computation of track index (TI) for quantifying image contrast.

| Enhancement Measure | Computation of Track Index | | | | | | | | | |
|---|---|---|---|---|---|---|---|---|---|---|
| | Off-Track Mean | On-Track Mean | Off-Track Sigma | On-Track Sigma | Off-Track CV | On-Track CV | TI-Diff | TI Ratio | TI-Ratio-Normalized | TI-CV Diff |
| Entropy | 0.91 | 0.04 | 0.14 | 0.10 | 0.16 | 2.52 | 0.87 | 2243.73 | 91.47 | 89.42 |
| Contrast | 0.34 | 0.00 | 0.25 | 0.00 | 0.73 | 2.27 | 0.34 | 131,279.4 | 99.85 | 80.38 |
| Energy | 0.03 | 0.94 | 0.05 | 0.15 | 1.66 | 0.16 | 0.91 | 3.15 | 93.89 | 53.37 |
| Homogeneity | 0.10 | 0.97 | 0.15 | 0.09 | 1.49 | 0.10 | 0.87 | 10.61 | 80.81 | 23.15 |
| Gray_Max | 0.71 | 1.00 | 0.21 | 0.00 | 0.30 | 0.00 | 0.29 | 70.87 | 17.05 | 100.00 |
| Gray | 0.52 | 1.00 | 0.32 | 0.01 | 0.61 | 0.01 | 0.48 | 51.80 | 31.76 | 92.15 |
| High-Pass Filter | 0.50 | 0.03 | 0.33 | 0.02 | 0.65 | 0.54 | 0.47 | 1453.30 | 87.12 | 24.58 |
| Laplacian | 0.50 | 0.51 | 0.35 | 0.02 | 0.70 | 0.04 | 0.00 | 99.08 | 0.46 | 79.81 |
| SobelE | 0.47 | 0.47 | 0.30 | 0.05 | 0.65 | 0.10 | 0.00 | 99.97 | 0.01 | 51.98 |

**Table 2.** Computation of effectiveness of different track indices.

| Enhancement Measure | Visual vs. Computed Rank | | | | | Difference from Actual (>2 Ranks) | | | |
|---|---|---|---|---|---|---|---|---|---|
| | Vision-Based Ranking | TI-Diff Rank | TI-Ratio Rank | TI-Ratio-Normalized Rank | TI-CV-Diff Rank | TI-Diff Rank | TI-Ratio Rank | TI-Ratio-Normalized Rank | TI-CV-Diff Rank |
| Entropy | 1 | 2 | 3 | 3 | 3 | 0 | 0 | 0 | 0 |
| Contrast | 2 | 6 | 1 | 1 | 4 | 1 | 0 | 0 | 0 |
| Energy | 3 | 1 | 2 | 2 | 7 | 0 | 0 | 0 | 1 |
| Homogeneity | 4 | 3 | 5 | 5 | 10 | 0 | 0 | 0 | 1 |
| Gray_Max | 5 | 7 | 7 | 7 | 1 | 0 | 0 | 0 | 1 |
| Gray | 6 | 4 | 6 | 6 | 2 | 0 | 0 | 0 | 1 |
| High-Pass Filter | 7 | 5 | 4 | 4 | 11 | 0 | 1 | 1 | 1 |
| Laplacian | 8 | 8 | 8 | 8 | 5 | 0 | 0 | 0 | 1 |
| SobelE | 9 | 9 | 9 | 9 | 8 | 0 | 0 | 0 | 0 |
| | % Accuracy | | | | | 88.9 | 88.9 | 88.9 | 33.3 |

## 5. Discussion

The study about enhancing the track contrast using various contrast enhancement measures as consolidated in various figures and tables demonstrates some important inferences. In the areas with low contrast, posing difficulty in the delineation of track areas, various image processing techniques [12,13] can be used to complement the interpretation process. This can help vision-systems-based decision-making processes to be more robust in their interpretation.

It is observed from Figures 1 and 2 that when a group of pixels representing any feature has a differentiable spatial arrangement from its surroundings, the role of texture assumes importance. The visual appearance of the contrast images demonstrates the role of texture in enhancing the track contrast in situations posing difficulty in the delineation of tracks.

The contrast images consolidated in Figures 1–3 indicate the role of GLCM in representing texture. Various statistic measures used here are seen as good indicators for delineating vehicle tracks. This supports the view expressed in [15] that GLCM-based measures give very good results in many fields of applications.

Depending upon the surrounding features, the most optimal texture measures enhancing track features are seen to vary. The visual appearance of different contrast images in Figure 6 demonstrates the varying role of different contrast enhancement measures in a given scenario. In order to make a comparative analysis, a quantitative method was proposed in this paper. The pixel values computed at various locations across the track were used for this purpose. The mean and standard deviation of the pixel values on-track and located off-track were also computed and normalized to facilitate a comparison of all images with different ranges of contrast enhancement measures.

Different forms of track indices have been explored and presented in this paper, as shown in Table 1. In order to compare the effectiveness of the track indices, the visual comparison was taken as the basis and the images were sorted based on each of the track indexes. The outcome of the comparison with visual perception as given in Table 2 demonstrates the effectiveness of the proposed track indices. The accuracy levels of the sorted images indicate that the proposed track-index-based approach is quite effective in sorting the contrast images based on the levels of contrast.

The proposed track-index-based approach can therefore be used for sorting the images of different enhancement measures based on track contrast. Although track indexes based on difference and ratio are both effective in sorting the images correctly, it is important that the influence of terrain features be considered here. For instance, in using the ratio-based track index, the Sobel- and Laplacian-based images, which have minimal variations in

mean values of on-track and off-track areas, could lead to unexpected results. This kind of issue capped the overall accuracy of the results using the proposed track index, which in the present case could result in 88% accuracy. A separate study could shed more light on better understanding the influence of varying topography, the size of the kernel, the width of the interpretation channel, etc. However, on a comparative basis, the track index based on the normalized ratio of difference is suggested to be used on preference, as it normalizes the comparison of different statistical measures rather than considering the absolute difference or ratio of the measures. An additional study could make this aspect even clearer for improving the result further. At this stage, machine learning tools could also be used to improve the accuracy levels of sorting the images even further based on inputs from all of these track indices.

While computing the track index, certain aspects need to be considered. The boundary areas of features in the track area influence the values of some enhancement measures, particularly in texture images. Therefore, for the computation of the track index, points in the track zone may thus be selected away from boundary areas for better inference of the track index.

Another study was carried out by the authors in desertic terrain to understand the influence of surrounding terrain features. Here, some additional parameters of GLCM were also considered and were related to GLCM mean and GLCM variance. The results shown in Figure 3 indicate that the track could be delineated better than the original image by using one or another image enhancement measure as per the details given by the investigators of [34,35]. However, the most optimal measure in a given situation depends upon the surrounding terrain features. Therefore, the procedure given in Figure 6 was adopted, which accounts for these variations and highlights the image with maximum track contrast.

Nowadays, machine learning is replacing human efforts. A number of attempts are being made in related studies of lane detection and its conditions, applying deep learning models. The accuracy of classification depends upon the accuracy and robustness of training sets. There are many cases when the vehicle tracks are there with feeble boundaries. This is where the proposed study of maximizing the track contrast could also be used for generating a robust and accurate dataset of track contrast.

The proposed study is seen to give a new method for making various tracks following off-road operations sustainable by improving decisions in low-contrast areas. This meets the requirements of both manual and autonomous navigation. Thus, the earlier works on the rut or track following vehicles as presented by [3,4,23] can be improved even further. The proposed methodology can support making intelligent on-board decisions for delineating track zones.

## 6. Conclusions

Using visual analytics, image intelligence and the optimal delineation of track features, a track-index-based contrast enhancement study was presented in this paper leading to the following set of key conclusions:

1.  In the locations with low contrast posing difficulty in the delineation of track areas, various image processing techniques can be used to complement the interpretation process.
2.  In the context of identifying track zones with a significant dimension, texture measures play a vital role in enhancing track contrast for the improved delineation of vehicle tracks.
3.  A track-index-based quantitative measure could effectively be used for the comparative analysis of different contrast enhancement measures with a wide range of variations.
4.  The proposed track-index-based technique can be usefully employed to sort out various images on the basis of track contrast with 88% accuracy, as seen in the current case. Further study to understand the influence of varying topography, the size of the

kernel and the width of the interpretation channel, etc., could lead to an improvement in track index.

5. The proposed visual analytics and track-index-based approach leading to improved inference of track features could augment the decision-making process for improved autonomous decisions in low-contrast areas.

The proposed study is a novel way to make various tracks following off-road operations sustainable by improving decisions in low-contrast areas. The proposed methodology can support intelligent decisions in on-board vehicles for the better delineation of track zones.

**Author Contributions:** Conceptualization, M.K.K., S.K.S. and A.T.; methodology, M.K.K.; software, M.K.K.; validation, M.K.K.; formal analysis, M.K.K.; investigation, M.K.K.; resources, M.K.K.; data curation, M.K.K.; writing—original draft preparation, M.K.K.; writing—review and editing, A.T. and S.K.S.; visualization, M.K.K., S.K.S. and A.T.; supervision, S.K.S. and A.T.;. All authors have read and agreed to the published version of the manuscript.

**Funding:** This received no external funding.

**Institutional Review Board Statement:** Not applicable.

**Informed Consent Statement:** Not applicable.

**Data Availability Statement:** Not applicable.

**Conflicts of Interest:** The authors declare no conflict of interest.

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
