# Peer review of "Track-Index-Guided Sustainable Off-Road Operations Using Visual Analytics, Image Intelligence and Optimal Delineation of Track Features"

_sustainability, doi:10.3390/su15107914_

Round 1
Reviewer 1 Report
The authors propose a comparison technique based on trajectory indices, based on the effectiveness of various images in improving trajectory contrast. The paper is well analyzed. The research work is of some value and recommended for acceptance, but the following issues need to be addressed prior to publication.
1. The description of the work and contributions in the introduction is not clear, please discuss it carefully.
2. Add a description of the structure of the paper in the last section of the introduction.
3. the formatting seems to be wrong, do not leave a large blank space. Also check your formulas, I see a strange form here.
Author Response
Comments and Suggestions by Reviewer # 1
Overall Comments and Suggestions for Authors:
The authors propose a comparison technique based on trajectory indices, based on the effectiveness of various images in improving trajectory contrast. The paper is well analyzed. The research work is of some value and recommended for acceptance.
Response by authors:
- The author(s) are thankful for these encouraging remarks. These will boost further attempts to improvise and explore the utilization of proposed study in different operational applications.
Referees Comments No. 1: The following issues need to be addressed prior to publication. The description of the work and contribution is not clear, please discuss it carefully.
Response by authors:
- Thank you for this valuable suggestion to improve the reflection of actual work. Based on this suggestion, suitable changes have been made in the revised manuscript in the Abstract and Introduction section.
Referees Comments No. 2: Add a description of the structure of the paper in the last section of the Introduction.
Response by authors:
- The suggested changes have been made in the revised manuscript.
Referees Comments No. 3: The formatting seems to be wrong, do not leave a large blank space. Also check the formulas, I see a strange form here.
Response by authors:
- Thank you for this valuable suggestion to improve the quality of presented work. The corrections have been made in the revised manuscript.
Reviewer 2 Report
1. Abstract and Conclusion should be concise yet. But should give complete overview of the work and study.
Authors can use latest related works from reputed journals like IEEE/ACM Transactions, Elsevier, Inderscience, Springer, Taylor & Francis etc and write the references in proper format, from year 2022-2023. Like https://www.sciencedirect.com/science/article/pii/S001048252200974X
https://link.springer.com/chapter/10.1007/978-3-031-17576-3_6
https://link.springer.com/article/10.1007/s11042-022-13551-2
2. The authors seem to disregard or neglect some important finding in results that have been achieved in paper. So, elaborate and explain the results in more details.
3. Improve the results and discussion section in paragraph.
4. Mention the future scope of your present works.
5. Paper should be formatted as per the requirements of the journal.
6. As a conclusion, the technical content is good. Therefore, the contribution of this article is also satisfactory. I am accepting article with minor revision for publication in this journal.
Author Response
Comments and Suggestions by Reviewer # 2
Referees Comments No. 1(a): Abstract and Conclusion should be concise yet. But should give complete overview of the work and study.
Response by authors:
- Thank you for this valuable suggestion to improve the reflection of actual work. Based on this suggestion, suitable changes have been made in the revised manuscript in the Abstract and Conclusion section.
Referees Comments No. 1(b): Authors can make use of latest works from reputed journals like IEEE/ACM Transactions, Elsevier, Inderscience, Springer, Taylor & Francis etc. and write the references in proper format, from year 2022-2023. Like:
https://www.sciencedirect.com/science/article/pii/S0001048252200974X
https://link.springer.com/chapter/10.1007/978-3-031-17576-3_6
https://link.springer.com/article/10.1007/s11042-022-13551-2
Response by authors:
- The authors hereby acknowledge that a lot of research continues in the object detection and segmentation using advanced machine learning techniques. Numerous attempts also exist in employing these machine learning techniques to address the issues of road surface delineation, its condition monitoring and object detection. However, it is pertinent yet to mention here that in the current study, the context is slightly different. The proposed work focuses on evolving a novel way to augment the on-board decision support system using image analytics. In a given environmental scenario, the vehicle tracks in low contrast areas are shown to be enhanced in automated way using the proposed technique. Although machine learning aspect is very important for further process of track detection, even for these studies to accurately mark the vehicle tracks with feeble boundaries, a robust dataset is needed. A good training set with images duly labeled with correctly marked tracks is needed. The proposed technique can also assist in creating a labeled dataset wherein the tracks in images are enhanced better by using conventional and unconventional image texture based algorithms as used in this study.
- A few points are clarified further citing some references that support the role of traditional image processing and computer vision techniques too. In the context of 3D robot vision, Burchfiel and Konidaris , 2018 have shown that combining both linear subspace methods and deep convolutional prediction achieves improved performance along with several orders of magnitude faster runtime performance compared to the state of the art. Marcus et al., 2018 present ten concerns for deep learning, and suggest that deep learning must be supplemented by other techniques if we are to reach artificial general intelligence. In the study of microcirculation images, Helmy et.al., 2022 (https://www.sciencedirect.com/science/article/pii/S2666389922002732) reported the limitation of deep learning and thus proposed a hybrid model to strike balance between accuracy and speed by combining traditional computer vision algorithms and CNN.
- Nevertheless, the authors re-assessed the literature pertaining to the proposed study. The limited references connected directly with the proposed track index based image enhancement for off-road areas that could be traced are included in the revised manuscript. For instance, A framework for Combined Road Tracking for Paved Roads and Dirt Roads is given given by Forkel et.al., 2021 (https://ieeexplore.ieee.org/document/9575141). In this work, a CNN-based measurement utilizing the self-similarity of (dirt) road areas is shown to be tracked with a lookahead length of 25 m. In another related study, Shamsolmoali et.al, 2020 presented work on use of Generative Adversarial Network (GAN) for addressing the issue of extracting road boundaries in complex terrain scenarios.
- Some other relevant literature from reputed journals connected with this kind of study is also a part of presented work. For instance, the studies published by IEEE from various researchers like Mei et.al., 2018; Humeau-Heurtier, 2019 and Kumar et.al., 2020 whose relevant aspects are included in the manuscript.
- Keeping in view the foregoing, suitable changes are made in the revised manuscript. The reference style is also revisited and is improvised.
References as cited above:
- Burchfiel B, Konidaris G. Hybrid bayesian eigenobjects: Combining linear subspace and deep network methods for 3D robot vision. In2018 IEEE/RSJ International Conference on Intelligent Robots and Systems (IROS) 2018 Oct 1 (pp. 6843-6850). IEEE.
- Marcus G. Deep learning: A critical appraisal. arXiv preprint arXiv:1801.00631. 2018.
- Helmy M, Truong TT, Jul E, Ferreira P. Deep learning and computer vision techniques for microcirculation analysis: A review. Patterns. 2022 Dec 1:100641.
- Forkel B, Kallwies J, Wuensche HJ. Combined Road Tracking for Paved Roads and Dirt Roads: Framework and Image Measurements. In2021 IEEE Intelligent Vehicles Symposium (IV) 2021 Jul 11 (pp. 1326-1331). IEEE.
- Shamsolmoali P, Zareapoor M, Zhou H, Wang R, Yang J. Road segmentation for remote sensing images using adversarial spatial pyramid networks. IEEE Transactions on Geoscience and Remote Sensing. 2020 Aug 21;59(6):4673-88.
Referees Comments No. 2: The authors seem to disregard or neglect some important finding in results that have been achieved in paper. So, elaborate and explain the results in more details.
Response by authors:
- Thank you for this insightful comment to improvise the findings. The results section is amended accordingly.
Referees Comments No. 3: Improve the results and discussion section in paragraph.
Response by authors:
- In light of the suggestions in the previous paragraphs, the changes in results and discussion section are made in the revised manuscript.
Referees Comments No. 4: Mention the future scope of your present works.
Response by authors:
- Identifying the limitations of the proposed approach and the possible extensions of the proposed study, the future scope section has been added. Further, since the proposed track index based study can not only enhance the dual vehicle tracks of leading vehicles but also the single tracks / linear road feature in the The track index is therefore amended slightly to cover this aspect too. The revised manuscript incorporates this change.
Referees Comments No. 5: Paper should be formatted as per the requirements of the journal.
Response by authors:
- The recommended guidelines for author(s) have been taken as the basis and the manuscript is suitably amended.
Referees Comments No. 6: As a conclusion, the technical content is good. Therefore, the contribution of this article is satisfactory. I am accepting article with minor revision for publication in this journal.
Response by authors:
- Thank you for the encouraging comments. This will be motivational for all the upcoming efforts to take this study further and fill the gaps to take it to operational use. The authors are really thankful for the thought provoking ideas given by the reviewer. These will remain in the background for the new initiatives to be taken in right direction.
Reviewer 3 Report
1. It is suggested to give the specific meaning of “c” in formula (2);
2. Formula (10) shall be modified to make it fully displayed;
3. Adjust the width of Figures 1-6 to make them consistent with the width of text;
4. Some references are old, and it is suggested to update them;
5. In order to verify the superiority of the proposed track index, the proposed track index should be compared with the existing track index.
Author Response
Comments and Suggestions by Reviewer # 3
Referees Comments No. 1: It is suggested to give the specific meaning of ‘c’ in formula (2).
Response by authors:
- The authors are really thankful for pointing such missed out points which could have been reflecting poor quality of presented work. Necessary details have been added in the revised manuscript.
Referees Comments No. 2: Formula (10) shall be modified to make it fully displayed.
Response by authors:
- The suggested correction has been made in the revised manuscript.
Referees Comments No. 3: Adjust the width of Figures 1-6 to make them consistent with the width of text.
Response by authors:
- The suggested correction has been made in the revised manuscript.
Referees Comments No. 4: Some references are old, and it is suggested to update them.
Response by authors:
- Some new references found relevant to this study have been identified and added in the revised manuscript.
Referees Comments No. 5: In order to verify the superiority of the proposed track index, the proposed index should be compared with the existing track index.
Response by authors:
- The literature is re-visited and no such work could be traced. However, this suggestion was explored in alternate way. Different forms of the proposed track index are worked out and proposed in this study as per the details given below:
- Based on Difference of Mean Values
The track index is defined on the basis of difference in mean values of pixels on-track and located Off-track. Depending upon the used statistic measure, the values of measure could be higher either on-track or off-track, the absolute difference is thus taken as the measure as described in the revised manuscript.
- Based on Ratio of Mean Values
The track index here is considered on the basis of ratio of the mean values of pixels on-track and located Off-track. Depending upon the used statistic measure, the values of measure could be higher either on-track or off-track, therefore the ratio of maximum and minimum values is taken as the measure as defined as described in the revised manuscript.
- Based on Normalised Differnce of Mean Values
The track index here is considered by normalizing the difference of mean values of pixels on-track and located Off-track. The measure as described in the revised manuscript.
- Based on the Ratio of Coefficient of Variance
The track index here considers the distinguishing feature of the track based on the standard deviation of values of pixels on-track and located Off-track. The Co-efficient of Variance (CV) which is the ratio of the standard deviation to the mean value is used here. Considering this measure, the variance based index is then defined as as described in the revised manuscript.
- All the above track indices are evaluated and compared for evaluating the effectiveness of proposed track index in this study. The revised manuscript presents the computed results and compares the proposed track index with other alternatives.
- The authors are really thankful for this thought provoking comment by the reviewer.

Reviewer 4 Report
After a careful review of the paper, I am suggesting the following comments to the authors to revise the manuscript.
1. Line 57: “There are several techniques…”. It is recommended to list several specific technologies.
2. Line 125: “Several applications make use of texture information …”. It is advised to provide examples or add references.
3. The formula formats are not standard, and there is no marked reference.
4. What does Y stand for in expression (3)? and what do n1 , n2 ,… stand for in expression (14)?
5. The (a), (b),... should be add in Figure 3, and what each picture represents should be explained. What the top half of Figure 5 represents should be explained.
6. Line 452: How was the 88% accuracy rate achieved?
7. Line 462: “In another study…”. What specific research has been done? What are the findings? Results should be given or references added.
8. When the track in the original gray image is difficult to identify, can the method in this paper be used to make the track more distinguishable?The methods used in this paper are all proposed by predecessors. Is there any improvement and innovation by the authors?
Author Response
Comments and Suggestions by Reviewer # 4
Referees Comments No. 1: Line 57: “There are several techniques…”. It is recommended to list several specific technologies.
Response by authors:
- The authors are really thankful for pointing out that would improve the quality of the work. The suggested changes have been made in the revised manuscript.
Referees Comments No. 2: Line 125: “Several applications make use of texture information… “. It is advised to provide examples or add references.
Response by authors:
- The suggested changes have been made in the revised manuscript.
Referees Comments No. 3: The formula formats are not standard and there is no marked reference.
Response by authors:
- The guidelines for authors have been re-visited and the suggested changes for improvement of the presented work have been made in the revised manuscript.
Referees Comments No. 4: What does γ stand for in expression (3)? And what do n1,n2… stand for in expression (14)?
Response by authors:
- The authors are really thankful for pointing such overlooked points which would have been reflecting poor on the quality of presented work. The suggested changes have been made in the revised manuscript.
Referees Comments No. 5: The (a), (b),…. Should be added in Figure 3, and what each picture represents should be explained. What the top half of Figure 5 represents should be explained.
Response by authors:
- The suggested changes have been made in the revised manuscript.
Referees Comments No. 6: Line 452: How was 88% accuracy rate achieved?
Response by authors:
- The accuracy level is computed by comparing contrast level of various images enhanced by different measures with the visual appearance of the relative track contrast.
- The proposed procedure has been amplified in further details in the revised manuscript. The underlying concept have been incorporated in the revised manuscript.
- In order to arrive at the accuracy levels of the sorted track images, the images enhanced for track contrast by different measures are first sorted based on visual interpretation.
- In order to compute the track contrast, the pixel values are computed across the track area. The mean and standard deviation of the pixel values on-track and those located off-track are then computed. Further computations are then carried out for computing the proposed track index. The details are shown in Table 1 of the revised manuscript.
- These indices indicate the comparative difference of track index values for different image enhancement measures. In order to evaluate the suitability of different measures, the images are sorted and ranked according to the track index values. The details as obtained are summarized in Table 2.
- The grading of each of the proposed track index is then compared with the visually graded rank of the images as per the track contrast. As the manually graded rank is subjected to some uncertainty in correct ordering, the difference of 2 levels in the ranks from the computed values are considered acceptable. The Table 2 indicates the results of the acceptable image ranks. The correctly marked images are then taken as the basis to evaluate the accuracy level which is given in the Table 2 of revised manuscript.
- The authors are thankful to the reviewer for making a note of this aspect as the novelty behind this procedure would have gone unnoticed.
Referees Comments No. 7: Line 462: “In another study …”, what specific research has been done? What are the findings? Results should be given or references added.
Response by authors:
- The intent here was to describe about another field study carried out by the authors in different kind of terrain. Suitable changes in the manuscript are made to clarify this aspect.
Referees Comments No. 8: When the track in original gray image is difficult to identify can the method in this paper be useful to make the track more distinguishable? The methods used in this paper are all proposed by predecessors. Is there any improvement and innovation by the authors?
Response by authors:
- It is quite common that the tracks of leading vehicles pass through low contrast areas which can cause difficulty in distinguishing the track contrast. In such a scenario, certain image processing techniques can assist in enhancing the contrast with reference to its surroundings. Moreover, since the tracks are of significant width as compared to camera resolution, the role of texture too comes into play which can assist in distinguishing these tracks based on spatial relation of pixels. It is true that these are standard techniques which are now re-drafted in the revised manuscript to convey the underlying message in better way.
- The essence of the proposed study is however different. As the quantum of track contrast varies as per the enhancement technique and also the surrounding terrain, the optimal selection of measure is targeted here. In this paper, a track index based comparative analysis is proposed which assists in sorting various images as per their effectiveness in increasing track contrast. The proposed technique is novel in the sense that it can select the optimal image in automated way for the decision makers in making the correct interpretation of vehicle tracks.
Considering the points as advised by reviewers, suitable changes have been made in the manuscript. Moreover, during the critical review of the manuscript, some points needing re-projecting the findings were also identified. These are also corrected wherever found in the revised manuscript. All the changes in the revised manuscript are highlighted with yellow colour. It is hoped that the revised manuscript addresses satisfactorily the points given by reviewers to improve the manuscript.
Round 2
Reviewer 4 Report
Thanks for your response.
I have seen that the relevant questions in the manuscript have been revised.